# GAR: Generalized Autoregression
# for Multi-Fidelity Fusion

**Yuxin Wang**[*]
School of Mathematical Science
Beihang University
Beijing, China, 100191.
`WYXtt_2011@163.com`

**Zheng Xing**[*]
Graphics&Computing Department
Rockchip Electronics Co., Ltd
Fuzhou, China, 350003
`zheng.xing@rock-chips.com`

**Wei W. Xing**[†]
School of Mathematics and Statistics, University of Sheffield, Sheffield S10 2TN, UK[‡]
School of Integrated Circuit Science and Engineering, Beihang University, Beijing, China, 100191.
`wayne.xingle@gmail.com`

## Abstract

In many scientific research and engineering applications where repeated simulations of complex systems are conducted, a surrogate is commonly adopted to quickly estimate the whole system. To reduce the expensive cost of generating training examples, it has become a promising approach to combine the results of low-fidelity (fast but inaccurate) and high-fidelity (slow but accurate) simulations. Despite the fast developments of multi-fidelity fusion techniques, most existing methods require particular data structures and do not scale well to high-dimensional output. To resolve these issues, we generalize the classic autoregression (AR), which is wildly used due to its simplicity, robustness, accuracy, and tractability, and propose generalized autoregression (GAR) using tensor formulation and latent features. GAR can deal with arbitrary dimensional outputs and arbitrary multi-fidelity data structure to satisfy the demand of multi-fidelity fusion for complex problems; it admits a fully tractable likelihood and posterior requiring no approximate inference and scales well to high-dimensional problems. Furthermore, we prove the autokrigeability theorem based on GAR in the multi-fidelity case and develop CIGAR, a simplified GAR with the exact predictive mean accuracy with computation reduction by a factor of $d^3$, where $d$ is the dimensionality of the output. The empirical assessment includes many canonical PDEs and real scientific examples and demonstrates that the proposed method consistently outperforms the SOTA methods with a large margin (up to 6x improvement in RMSE) with only a couple high-fidelity training samples.

## 1 Introduction

The design, optimization, and control of many systems in science and engineering can rely heavily on the numerical analysis of differential equations, which is generally computationally intense. In this case, a data-driven surrogate model is used to approximate the system based on the input-output data of the numerical solver and to help improve convergence efficiency where repeated simulations are required, e.g., in Bayesian optimization (BO) [1] and uncertainty analysis (UQ) [2].

With the surrogate model in place, the remaining challenge is that executing high-fidelity numerical simulations to generate training data can still be very expensive. To further reduce the computational

---

[*]The authors contribute equally to this paper.  [†]Corresponding author.  [‡]Primary

36th Conference on Neural Information Processing Systems (NeurIPS 2022).

burden, it is possible to combine low-fidelity results to make high-fidelity predictions [3]. More specifically, low-fidelity solvers are normally based on simplified PDEs (e.g., reducing the levels of physical detail) or simple solver setup (e.g., using a coarse mesh, a large time step, a lower order of an approximating basis, and a higher error tolerance). They provide fast but inaccurate solutions to the original problems whereas the high-fidelity solvers are accurate yet expensive. The multi-fidelity fusion technique works similar to transfer learning to utilize many low-fidelity observations to improve the accuracy when using only a few high-fidelity samples. In general, it involves constructions of surrogates for different fidelities and a cross-fidelity transition process. Due to its efficiency, multi-fidelity method has attracted increasing attention in BO [4, 5], UQ [6, 7], and surrogate modeling [8, 9]. We refer to [10, 11] for a great review.

Despite the success of many state-of-the-art (SOTA) approaches, they normally assume that (1) the output dimension is the same and aligned across all fidelities, which generally does not hold for multi-fidelity simulation where the output are quantities at nodes that are naturally not aligned; (2) the high-fidelity samples' corresponding inputs form a subset of the low-fidelity inputs; and (3) the output dimension is small, which is not practical for scientific computing where dimension can be 1-million (for a $100 \times 100 \times 100$ spatial-temporal field). These assumptions seriously hinder their applications for practical problems, e.g., MRI imaging and solving PDEs in scientific computation.

To resolve these challenges, previous work either uses interpolation to align the dimension [12, 9] or relies on approximate inference with brutal simplification [8, 13], leading to inferior performance. We notice that the classic autoregression (AR), which is widely used due to its simplicity, robustness, accuracy, and tractability, consistently shows robust and top-tier performance for different datasets in the literature, despite its incapability for high-dimensional problems. Thus, instead of proposing another ad-hoc model with pre-processing and simplification (leading to models that are difficult to tune and generalize poorly), we generalize AR with tensor algebra and latent features and propose generalized autoregression (GAR), which can deal with arbitrary high-dimensional problems without the subset multi-fidelity data structure. GAR is a fully tractable Bayesian model with scalability to extremely high-dimensional outputs, without requiring any approximate inference. The novelty of this work is as follows,

1. Generalization of AR for arbitrary non-structured and high-dimensional outputs. With tensor algebra and latent features, GAR allows effective knowledge transfer in closed-form and is scalable to extreme high-dimensional problems.

2. Generalization to non-subset multi-fidelity data for AR. To the best of our knowledge, we are the first to generalize the closed form solution of subset data to non-subset cases for AR and the proposed GAR.

3. For the first time, we reveal the autokrigeability [14] for the multi-fidelity fusion within an AR structure, based on which we derive conditional independent GAR (CIGAR), an efficient implementation of GAR with the exact accuracy in posterior mean predictions.

## 2   Backgronud

### 2.1   Statement of the problem

Given multi-fidelity data $D^i = \{\mathbf{x}_n^i, \mathbf{y}_n^i\}_{n=1}^{N^i}$, for $i = 1, \ldots, \tau$, where $\mathbf{x} \in \mathbb{R}^l$ denotes the system inputs (a vector of parameters that appear in the system of equations and/or in the initial-boundary conditions for a simulation); $\mathbf{y}^i \in \mathbb{R}^{d^i}$ denotes the corresponding outputs, where $d^i$ is the dimension for $i$ fidelity; $\tau$ is the total number of fidelities. Generally speaking, higher-fidelity data are closer to the ground truth and are more expensive to obtain. Thus, we have fewer samples for the high fidelity, i.e., $N^1 \geqslant N^2 \geqslant \cdots \geqslant N^\tau$. The dimensionality is not necessary the same or aligned across different fidelities. In most work e.g., [15, 16, 11], the system inputs of higher-fidelity are chosen to be the subset of the lower-fidelity, i.e., $\mathbf{X}^\tau \subset \cdots \subset \mathbf{X}^2 \subset \mathbf{X}^1$. We call this the subset structure for a multi-fidelity dataset, as opposed to arbitrary data structures, which we will resolve in Section 3.3 with a closed-form solution and extend it to the classic AR. Our goal is to estimate the function $\mathbf{f}^\tau : \mathbb{R}^l \to \mathbb{R}^{d^\tau}$ given the observation data across different fidelities $\{D^i\}_{i=1}^\tau$.

## 2.2 Autoregression

For the sake of clarity, we consider a two-fidelity case with superscript $h$ indicating high-fidelity and $l$ low-fidelity. Nevertheless, the formulation can be generalized to problems with multiple fidelities recursively. Considering a simple scalar output for all fidelities, AR [3] assumes

$$f^h(\mathbf{x}) = \rho f^l(\mathbf{x}) + f^r(\mathbf{x}), \tag{1}$$

where $\rho$ is a factor transferring knowledge from the low fidelity in a linear fashion, whereas $f^r(\mathbf{x})$ tries to capture the residual information. If we assume a zero mean Gaussian process (GP) prior [17] (see Appendix A for a brief description) for $f^l(\mathbf{x})$ and $f^r(\mathbf{x})$, i.e., $f^l(\mathbf{x}) \sim \mathcal{N}(0, k^l(\mathbf{x}, \mathbf{x}'))$ and $f^r(\mathbf{x}) \sim \mathcal{N}(0, k^r(\mathbf{x}, \mathbf{x}'))$, the high-fidelity function also follows a GP. This gives an elegant joint GP for the joint observations $\mathbf{y} = [\mathbf{y}^l; \mathbf{y}^h]^T$,

$$\begin{pmatrix} \mathbf{y}^l \\ \mathbf{y}^h \end{pmatrix} \sim \mathcal{N}\left( \mathbf{0}, \begin{matrix} \mathbf{K}^l(\mathbf{X}^l, \mathbf{X}^l) & \rho\mathbf{K}^l(\mathbf{X}^l, \mathbf{X}^h) \\ \rho\mathbf{K}^l(\mathbf{X}^h, \mathbf{X}^l) & \rho^2\mathbf{K}^l(\mathbf{X}^h, \mathbf{X}^h) + \mathbf{K}^r(\mathbf{X}^h, \mathbf{X}^h) \end{matrix} \right) \tag{2}$$

where $\mathbf{y}^l \in \mathbb{R}^{N_l \times 1}$ is the low-fidelity observations corresponding to input $\mathbf{X}^l \in \mathbb{R}^{N_l \times L}$ and $\mathbf{y}^h \in \mathbb{R}^{N_h \times 1}$ is the high-fidelity observations; $[\mathbf{K}^l(\mathbf{X}^l, \mathbf{X}^l)]_{ij} = k^l(\mathbf{x}_i, \mathbf{x}_j)$ is the covariance matrix of the low-fidelity inputs $\mathbf{x}_i, \mathbf{x}_j \in \mathbf{X}^l$; $[\mathbf{K}^r(\mathbf{X}^l, \mathbf{X}^l)]_{ij} = k^r(\mathbf{x}_i, \mathbf{x}_j)$ is for the high-fidelity inputs $\mathbf{x}_i, \mathbf{x}_j \in \mathbf{X}^h$; $[\mathbf{K}^l(\mathbf{X}^l, \mathbf{X}^h)]_{ij} = k^l(\mathbf{x}_i, \mathbf{x}_j)$ is the cross-fidelity covariance matrix of the low-fidelity inputs $\mathbf{x}_i \in \mathbf{X}^l$ and high-fidelity inputs $\mathbf{x}_j \in \mathbf{X}^h$, and $\mathbf{K}^l(\mathbf{X}^h, \mathbf{X}^l) = (\mathbf{K}^l(\mathbf{X}^l, \mathbf{X}^h))^T$. One immediate advantage of AR is that the joint Gaussian form allows not only joint training for all low- and high-fidelity data but also predictions for any given new inputs by a conditional Gaussian (as the posterior is derived in a standard GP [17]). Furthermore, Le Gratiet [15] derive Lemma 1 to reduce the complexity from $O((N^l + N^h)^3)$ to $O((N^l)^3 + (N^h)^3)$ with a subset data structure.

**Lemma 1.** *[15] If $\mathbf{X}^h \subset \mathbf{X}^l$, the joint likelihood and predictive posterior of AR can be decomposed into two independent parts corresponding to the low- and high-fidelity data.*

# 3 Generalized Autoregression

Let's now consider the more general high-dimensional case. A naive approach is to simply convert the multi-dimensional output into a scalar value by attaching a dimension index to the input. However, AR will end up with a joint GP with a covariance matrix of the size of $(N^l d^l + N^h d^h)^2$, making it infeasible for modestly high-dimensional problems.

## 3.1 Tensor Factorized Generalization with Latent Features

To resolve the scalability issue, we rearrange all the output into a multidimensional space (i.e., a tensor space) and introduce latent coordinate features to index the outputs to capture their correlations as in HOGP [18]. More specifically, we organize the low-fidelity output as a $M$-mode tensor, $\mathbf{Z}^l \in \mathbb{R}^{d_1^l \times \cdots \times d_M^l}$, where the output dimension $d^l = \prod_{m=1}^M d_m^l$. The element $\mathbf{Z}^l$ is indexed based on its coordinates $\mathbf{c} = (c_1, \ldots, c_M)(1 \leqslant c_k \leqslant d_k$ for $k = 1, \ldots, M)$. If the original data indeed admits a multi-array structure, we can use their original index with actual meaning to index the coordinates. For instance, a 2D spatial dataset can use its original spatial coordinate to index a single location (pixel). To improve our model flexibility, we do not have to limit ourselves from using the original index, particularly for the cases where the original data does not admit a multi-array structure or the multi-array structure is of too large size. In such case, we can use arbitrary tensorization and a latent feature vector $\mathbf{v}_{c_m}^l$ (whose values are inferred in model training) for each coordinate $c_m$ in mode $m$. This way, the element $\mathbf{Z}^l$ is indexed by the vector $(\mathbf{v}_{c_1}^l, \ldots, \mathbf{v}_{c_M}^l)$. Following the linear transformation of Eq. (1), we first introduce the tensor-matrix product [19],

$$\mathbf{F}^h(\mathbf{x}) = \mathbf{F}^l(\mathbf{x}) \times_1 \mathbf{W}_1, \ldots, \times_M \mathbf{W}_M + \mathbf{F}^r(\mathbf{x}), \tag{3}$$

where $\mathbf{F}^h(\mathbf{x})$ denotes target function $\mathbf{f}^h(\mathbf{x})$ with its output organized into a multi-array $\mathbf{Z}^h$, and the same concept applies to $\mathbf{F}^l(\mathbf{x})$ and $\mathbf{F}^r(\mathbf{x})$; $\times_m$ denotes the tensor-matrix product at mode $m$. To give a concrete example, considering an arbitrary tensor $\mathbf{Z}^l \in \mathbb{R}^{d_1^l \times \cdots \times d_M^l}$ and a matrix $\mathbf{W}_m \in \mathbb{R}^{s \times d_m}$, the $\times_m$ product is calculated as $[\mathbf{Z}^l \times_m \mathbf{W}_m]_{i_1, \ldots, i_{m-1}, j, i_{m+1} \ldots, i_M} = \sum_{k=1}^{c_m} w_{jk} Z_{i_1, \ldots, i_{m-1}, k, i_{m+1} \ldots, i_M}$, which becomes an $d_1^l \times \cdots \times d_{m-1}^l \times s \times d_{m+1}^l \times \cdots \times d_M^l$ tensor. We can further denote the group

of $M$ linear transformation matrixes as a Tucker tensor $\mathbf{W} = [\mathbf{W}_1, \ldots, \mathbf{W}_M]$ and represent Eq. (3) compactly using a Tucker operator [19], $\mathbf{F}^l(\mathbf{x}) \times \mathbf{W}$, which has an important property:

$$\text{vec}\left(\mathbf{F}^h(\mathbf{x}) - \mathbf{F}^r(\mathbf{x})\right) = (\mathbf{W}_1 \otimes \ldots \otimes \mathbf{W}_M)\,\text{vec}\left(\mathbf{F}^l(\mathbf{x})\right). \tag{4}$$

Inspired by AR of Eq. (1), we place a tensor-variate GP (TGP) prior [20] for the low-fidelity tensor function $\mathbf{F}^l(\mathbf{x})$ and the residual tensor function $\mathbf{F}^r(\mathbf{x})$:

$$\mathbf{Z}^l(\mathbf{x}, \mathbf{x}') \sim \mathcal{TGP}\left(\mathbf{0}, k^l(\mathbf{x}, \mathbf{x}'), \mathbf{S}_1^l, \ldots, \mathbf{S}_M^l\right), \mathbf{Z}^r(\mathbf{x}, \mathbf{x}') \sim \mathcal{TGP}\left(\mathbf{0}, k^r(\mathbf{x}, \mathbf{x}'), \mathbf{S}_1^r, \ldots, \mathbf{S}_M^r\right), \tag{5}$$

where $\mathbf{S}_m^i \in \mathbb{R}^{d_m \times d_m}$ are the output correlation matrix with $[\mathbf{S}_m^i]_{jk} = \tilde{k}_m^i(\mathbf{v}_{c_i}^i, \mathbf{v}_{c_k}^i)$ and $\tilde{k}_m^i(\cdot, \cdot)$ being the kernel function (with unknown hyperparameters). A TGP is a generalization of a multivariate GP that essentially represents a joint GP prior $\text{vec}(\mathbf{Y}^l) \sim \mathcal{N}\left(0, \mathbf{K}^l(\mathbf{X}^l, \mathbf{X}^l) \bigotimes_{m=1}^M \mathbf{S}_m\right)$. Similar to the joint probability of (2), we can derive the joint probability for $\mathbf{y} = [\text{vec}(\mathbf{Y}^l)^T, \text{vec}(\mathbf{Y}^h)^T]^T$ based on Tucker transformation of (3); we preserve the proof in the Appendix for clarity.

**Lemma 2.** *Given the tensor GP priors for $\mathbf{Y}^l(\mathbf{x}, \mathbf{x}')$ and $\mathbf{Y}^r(\mathbf{x}, \mathbf{x}')$ and the Tucker transformation of* (3), *the joint probability for $\mathbf{y} = [\text{vec}(\mathbf{Y}^l)^T, \text{vec}(\mathbf{Y}^h)^T]^T$ is $\mathbf{y} \sim \mathcal{N}(\mathbf{0}, \mathbf{\Sigma})$, where $\mathbf{\Sigma} =$*

$$\begin{pmatrix} \mathbf{K}^l(\mathbf{X}^l, \mathbf{X}^l) \otimes \left(\bigotimes_{m=1}^M \mathbf{S}_m\right) & \mathbf{K}^l(\mathbf{X}^l, \mathbf{X}^h) \otimes \left(\bigotimes_{m=1}^M \mathbf{S}_m \mathbf{W}_m^T\right) \\ \mathbf{K}^l(\mathbf{X}^h, \mathbf{X}^l) \otimes \left(\bigotimes_{m=1}^M \mathbf{W}_m \mathbf{S}_m\right) & \mathbf{K}^l(\mathbf{X}^h, \mathbf{X}^h) \otimes \left(\bigotimes_{m=1}^M \mathbf{W}_m \mathbf{S}_m \mathbf{W}_m^T\right) + \mathbf{K}^r(\mathbf{X}^h, \mathbf{X}^h) \otimes \left(\bigotimes_{m=1}^M \mathbf{S}_m^r\right) \end{pmatrix}.$$

Lemma 2 admits any arbitrary outputs (living in different spaces, having different dimension and/or mode, and being unaligned) at different fidelity. Also, it does not require a subset dataset to hold.

**Corollary 3.0.1.** *Lemma 2 can be applied to data with a different number of mode at each fidelity, i.e., $M^l \neq M^h$, if we add a redundancy index such that all outputs have the same $M$ number of modes.*

Lemma 2 defines our GAR model, a generalized AR with special tensor structures. The covariance for low-fidelity is $\text{cov}(Z_{\mathbf{c}}^l(\mathbf{x}), Z_{\mathbf{c}'}^l(\mathbf{x}')) = k^l(\mathbf{x}, \mathbf{x}') \prod_{m=1}^M \tilde{k}_m^l(\mathbf{v}_{c_m}^m, \mathbf{v}_{c_m'}^m)$, cross-fidelity $\text{cov}(Z_{\mathbf{c}}^l(\mathbf{x}), Z_{\mathbf{c}'}^h(\mathbf{x}')) = k^l(\mathbf{x}, \mathbf{x}') \prod_{m=1}^M \tilde{k}_m^l(\mathbf{v}_{c_m}^l, \mathbf{v}_{c_m'}^l) w_{c,c'}^m$ (where $w_{c,c'}^m$ is the $c, c'$-th element of $\mathbf{W}^m$), and high-fidelity $\text{cov}(Z_{\mathbf{c}}^h(\mathbf{x}), Z_{\mathbf{c}'}^h(\mathbf{x}')) = k^l(\mathbf{x}, \mathbf{x}') \prod_{m=1}^M \tilde{k}_m^l(\mathbf{v}_{c_m}^l, \mathbf{v}_{c_m'}^l)(w_{c,c'}^m)^2 + k^h(\mathbf{x}, \mathbf{x}') \prod_{m=1}^M \tilde{k}_m^r(\mathbf{u}_{c_m}^m, \mathbf{u}_{c_m'}^m)(w_{c,c'}^m)^2$. The complex between-fidelity output correlations are captured using latent features $\{\mathbf{V}^m, \mathbf{U}^m\}_{m=1}^M$ with arbitrary kernel function $\tilde{k}_m^i$, whereas the cross-fidelity output correlations are captured in a composite manner. This combination overcomes the simple linear correlations assumed in previous work that simply decomposes the output as a dimension reduction preprocess [12]. When the dimensionality aligns for $\mathbf{Z}^l$ and $\mathbf{Z}^h$ and thus $d_m^l = d_m^h$, we can share the same latent features across the two fidelities by letting $\mathbf{v}_j^m = \mathbf{u}_j^m$ while keeping the kernel functions different. This way, the latent features are more resistant to overfitting. For non-aligned data with explicit indexing, we can use kernel interpolation [21] for the same purpose. To further encourage sparsity in the latent feature, we impose a Laplace prior, i.e., $\mathbf{v}_j^m \sim \text{Laplace}(\lambda) \propto \exp(-\lambda||\mathbf{v}_j^m||_1)$.

### 3.2 Efficient Model Inference for Subset Data Structure

With the model fully defined, we can now train the model to obtain all unknown model parameters. For compactness, we use the following compact notation $\mathbf{S}^l = \bigotimes_{m=1}^M \mathbf{S}_m^l, \mathbf{S}^h = \bigotimes_{m=1}^M \mathbf{S}_m^h, \mathbf{W} = \bigotimes_{m=1}^M \mathbf{W}_m, \mathbf{K}^l = \mathbf{K}^l(\mathbf{X}^l, \mathbf{X}^l), \mathbf{K}^{lh} = \mathbf{K}^l(\mathbf{X}^l, \mathbf{X}^h), \mathbf{K}^{hl} = \mathbf{K}^l(\mathbf{X}^h, \mathbf{X}^l), \mathbf{K}^{lr} = \mathbf{K}^l(\mathbf{X}^h, \mathbf{X}^h)$, and, $\mathbf{K}^r = \mathbf{K}^r(\mathbf{X}^h, \mathbf{X}^h)$ (with a slight abuse of notation).

**Lemma 3.** *Tensor generalization of Lemma 1. If $\mathbf{X}^h \subset \mathbf{X}^l$, the joint likelihood $\mathcal{L}$ for $\mathbf{y} = [\text{vec}(\mathbf{Y}^l)^T, \text{vec}(\mathbf{Y}^h)^T]^T$ admits two independent separable likelihoods $\mathcal{L} = \mathcal{L}^l + \mathcal{L}^r$, where*

$$\mathcal{L}^l = -\frac{1}{2}\text{vec}\left(\mathbf{Y}^l\right)^T (\mathbf{K}^l \otimes \mathbf{S}^l)^{-1}\text{vec}\left(\mathbf{Y}^l\right) - \frac{1}{2}\log|\mathbf{K}^l \otimes \mathbf{S}^l| - \frac{N^l D^l}{2}\log(2\pi),$$

$$\mathcal{L}^r = -\frac{1}{2}\text{vec}\left(\mathbf{Y}^h - \mathbf{Y}^l \times \hat{\mathbf{W}}\right)^T (\mathbf{K}^r \otimes \mathbf{S}^r)^{-1}\text{vec}\left(\mathbf{Y}^h - \mathbf{Y}^l \times \hat{\mathbf{W}}\right) - \frac{1}{2}\log|\mathbf{K}^r \otimes \mathbf{S}^r| - \frac{N^h D^h}{2}\log(2\pi),$$

*where $\hat{\mathbf{W}} = [\mathbf{E}, \mathbf{W}_1, \ldots, \mathbf{W}_M]$ is a Tucker tensor with selection matrix $\mathbf{E}^T \mathbf{X}^l = \mathbf{X}^h$.*

We preserve the proof in Appendix for clarity. Note that $\mathcal{L}^l$ and $\mathcal{L}^r$ are HOGP likelihoods for $\mathbf{Y}^l$ and the residual $\mathbf{Y}^h - \mathbf{Y}^l \times \hat{\mathbf{W}}$, respectively. Since the computational of $\mathcal{L}^l$ and $\mathcal{L}^r$ are independent, the model training can be conducted efficiently in parallel.

**Predictive posterior.** Similarly, we can derive the concrete predictive posterior for the high-fidelity outputs by integrating out the latent functions after some tedious linear algebra (see Appendix), which is also Gaussian, $\mathrm{vec}(\mathbf{Z}_*^h) \sim \mathcal{N}(\mathrm{vec}(\bar{\mathbf{Z}}_*^h), \mathbf{S}_*^h)$, where

$$
\begin{aligned}
\mathrm{vec}(\bar{\mathbf{Z}}_*^h) &= \left( \mathbf{k}_*^l \left(\mathbf{K}^l\right)^{-1} \otimes \mathbf{W} \right) \mathrm{vec}(\mathbf{Y}^l) + \left( \mathbf{k}_*^r \left(\mathbf{K}^r\right)^{-1} \otimes \mathbf{I} \right) \mathrm{vec}(\mathbf{Y}^r), \\
\mathbf{S}_*^h &= \left( k_{**}^l - (\mathbf{k}_*^l)^T \left(\mathbf{K}^l\right)^{-1} \mathbf{k}_*^l \right) \otimes \mathbf{W}\mathbf{S}^l\mathbf{W}^T + \left( k_{**}^r - (\mathbf{k}_*^r)^T \left(\mathbf{K}^r\right)^{-1}\mathbf{k}_*^r \right) \otimes \mathbf{S}^r,
\end{aligned}
\tag{6}
$$

$\mathbf{k}_*^l = \mathbf{k}^l(\mathbf{x}_*, \mathbf{X}^l)$ is the vector of covariance between the give input $\mathbf{x}_*$ and low-fidelity observation inputs $\mathbf{X}^l$; similarly, we have $\mathbf{k}_{**}^l = \mathbf{k}^l(\mathbf{x}_*, \mathbf{x}_*)$, $\mathbf{k}_*^r = \mathbf{k}^r(\mathbf{x}_*, \mathbf{X}^h)$, $\mathbf{k}_{**}^r = \mathbf{k}^r(\mathbf{x}_*, \mathbf{x}_*)$.

### 3.3 Generalization for Non-subset Data: Efficient Model Inference and Prediction

In practice, it is sometimes difficult to ask the multi-fidelity data to preserve a subset structure, particularly in the case of multi-fidelity Bayesian optimization [22, 23]. This presents the challenge for most SOTA multi-fidelity models e.g., NAR [16], ResGP [9], stochastic collocation [24]. In contrast, the advantage of AR is that even if the multi-fidelity data does not admit a subset data structure, the model can still be trained using all available data based on the joint likelihood of (2). However, this method lacks scalability due to the inversion of the large joint covariance matrix $\Sigma$. The situation gets worse if we are dealing with multi-fidelity data with more than two fidelities. To resolve this issue, we propose a fast inference method based on imaginary subset. More specifically, considering the missing low-fidelity data as latent variables $\hat{\mathbf{Y}}^l$, the joint likelihood function is

$$
\begin{aligned}
\log p(\mathbf{Y}^l, \mathbf{Y}^h) &= \log \int p(\mathbf{Y}^l, \mathbf{Y}^h, \hat{\mathbf{Y}}^l) d\hat{\mathbf{Y}}^l = \log \int \left( p(\mathbf{Y}^h | \hat{\mathbf{Y}}^l, \mathbf{Y}^l) p(\hat{\mathbf{Y}}^l | \mathbf{Y}^l) p(\mathbf{Y}^l) \right) d\hat{\mathbf{Y}}^l \\
&= \log \int p(\mathbf{Y}^h | \hat{\mathbf{Y}}^l, \mathbf{Y}^l) p(\hat{\mathbf{Y}}^l | \mathbf{Y}^l) d\hat{\mathbf{Y}}^l + \log p(\mathbf{Y}^l),
\end{aligned}
\tag{7}
$$

where $p(\mathbf{Y}^h | \hat{\mathbf{Y}}^l, \mathbf{Y}^l)$ is the likelihood in Lemma 3 given the complementary imaginary subset, and $p(\hat{\mathbf{Y}}^l | \mathbf{Y}^l) \sim \mathcal{N}(\bar{\mathbf{Y}}^l, \hat{\mathbf{S}}^l \otimes \mathbf{S}^l)$ is the imaginary posterior with the given low-fidelity observations $\mathbf{Y}^l$. The integral can be calculated using Gaussian quadrature or other sampling methods as in [8, 25], which is slow and inaccurate.

**Lemma 4.** *The joint likelihood of GAR for non-subset (and also unaligned) data can be decomposed into two independent GPs' likelihood*

$$
\log p(\mathbf{Y}^l, \mathbf{Y}^h) = \mathcal{L}^l - \frac{N^h d^h}{2}\log(2\pi) - \frac{1}{2}\log\left| \mathbf{K}^r \otimes \mathbf{S}^r + \hat{\mathbf{E}}\hat{\mathbf{S}}^l\hat{\mathbf{E}}^T \otimes \mathbf{W}^T\mathbf{S}^l\mathbf{W} \right|
$$
$$
- \frac{1}{2}\left[ \left( \begin{array}{c} \mathrm{vec}(\check{\mathbf{Y}}^h) \\ \mathrm{vec}(\hat{\mathbf{Y}}^h) \end{array} \right)^T - \left( \begin{array}{c} \mathrm{vec}(\check{\mathbf{Y}}^l) \\ \mathrm{vec}(\bar{\mathbf{Y}}^l) \end{array} \right)^T \tilde{\mathbf{W}}^T \right] \left( \mathbf{K}^r \otimes \mathbf{S}^r + \hat{\mathbf{E}}\hat{\mathbf{S}}^l\hat{\mathbf{E}}^T \otimes \mathbf{W}^T\mathbf{S}^l\mathbf{W} \right)^{-1} \left[ \left( \begin{array}{c} \mathrm{vec}(\check{\mathbf{Y}}^h) \\ \mathrm{vec}(\hat{\mathbf{Y}}^h) \end{array} \right) - \tilde{\mathbf{W}}\left( \begin{array}{c} \mathrm{vec}(\check{\mathbf{Y}}^l) \\ \mathrm{vec}(\bar{\mathbf{Y}}^l) \end{array} \right) \right],
\tag{8}
$$

*where $\mathcal{L}^l$ is the likelihood for low-fidelity data $\mathbf{Y}^l$, $\tilde{\mathbf{W}} = \mathbf{I}_{N^h} \otimes \mathbf{W}$ $\hat{\mathbf{Y}}^h$ is the collection of high-fidelity observations corresponding to the imaginary low-fidelity outputs $\hat{\mathbf{Y}}^l$; $\check{\mathbf{Y}}^h$ is the complement (with selection matrix $\check{\mathbf{X}}^h = \mathbf{E}^T\mathbf{X}^l$) corresponding to low-fidelity outputs $\check{\mathbf{Y}}^l$, i.e., $\mathbf{Y}^h = [\check{\mathbf{Y}}^h, \hat{\mathbf{Y}}^h]$; and $\hat{\mathbf{X}}^h = \hat{\mathbf{E}}^T\mathbf{X}^h$ are the selection matrix for $\hat{\mathbf{Y}}^l$.*

We preserve the proof in the appendix. Notice that $\hat{\mathbf{E}}\hat{\mathbf{S}}^l\hat{\mathbf{E}}^T \otimes \mathbf{W}^T\mathbf{S}^l\mathbf{W} = \left( \begin{array}{cc} \mathbf{0} & \mathbf{0} \\ \mathbf{0} & \hat{\mathbf{S}}^l \end{array} \right) \otimes \mathbf{W}^T\mathbf{S}^l\mathbf{W}$

is the low-right block of the predictive variance for the missing low-fidelity observations $\mathrm{vec}(\hat{\mathbf{Y}})$; We can easily understand the last part of the likelihood as a GP with accumulated uncertainty(variance) added to the corresponding missing points. Lemma 4 naturally applies to AR when the outputs is a scaler, where $\mathbf{W} = \rho$, $\mathbf{S}^l = 1$, and $\mathbf{S}^r = 1$.

**Predictive posterior.** Surprisingly, the posterior also turns out to be a Gaussian distribution,

$$p\left(\mathbf{Z}_*^h|\mathbf{Y}^l, \mathbf{Y}^h, \mathbf{x}_*\right) = 2\pi^{-\frac{d^h}{2}} \times \left|\mathbf{S}_*^h + \mathbf{\Gamma}\left(\hat{\mathbf{S}}^l \otimes \mathbf{S}^l\right)\mathbf{\Gamma}^T\right|^{-\frac{1}{2}}$$
$$\times \exp\left[-\frac{1}{2}\left(\mathrm{vec}(\mathbf{Z}_*^h) - \mathrm{vec}(\bar{\mathbf{Z}})\right)^T\left(\mathbf{S}_*^h + \mathbf{\Gamma}\left(\hat{\mathbf{S}}^l \otimes \mathbf{S}^l\right)\mathbf{\Gamma}^T\right)^{-1}\left(\mathrm{vec}(\mathbf{Z}_*^h) - \mathrm{vec}(\bar{\mathbf{Z}})\right)\right], \tag{9}$$

where $\mathbf{\Gamma}$ and the mean of the predictive posterior $\bar{\mathbf{Z}}_*$ are given as follows,

$$\mathbf{\Gamma} = \left([\mathbf{k}_*^r(\mathbf{K}^r)^{-1}\mathbf{E}_n^T - \mathbf{k}_*^l(\hat{\mathbf{K}}^l)^{-1}] \otimes \mathbf{W}\right)\mathbf{E}_m \otimes \mathbf{I}^l,$$

$$\mathrm{vec}(\bar{\mathbf{Z}}_*) = \left(\mathbf{k}_*^l(\hat{\mathbf{K}}^l)^{-1} \otimes \mathbf{W}\right)\begin{pmatrix}\mathrm{vec}(\mathbf{Y}^l)\\\mathrm{vec}(\bar{\mathbf{Y}}^l)\end{pmatrix} + \left(\mathbf{k}_*^r(\mathbf{K}^r)^{-1} \otimes \mathbf{I}\right)\left(\mathrm{vec}(\mathbf{Y}^h) - \hat{\mathbf{W}}\begin{pmatrix}\mathrm{vec}(\mathbf{Y}^l)\\\mathrm{vec}(\bar{\mathbf{Y}}^l)\end{pmatrix}\right),$$
$$\tag{10}$$

where $\mathbf{E}_m$ and $\mathbf{E}_n$ are the selection matrices such that $\hat{\mathbf{X}}^h = \mathbf{E}_m^T[\mathbf{X}^l, \hat{\mathbf{X}}^h]$, $\mathbf{X}^h = \mathbf{E}_n^T[\mathbf{X}^l, \hat{\mathbf{X}}^h]$, $\hat{\mathbf{W}} = \mathbf{E}_n^T \otimes \mathbf{W}$, and $\hat{\mathbf{K}}^l$ is the covariance matrix that $\hat{\mathbf{K}}^l = \mathbf{K}^l([\mathbf{X}^l, \hat{\mathbf{X}}^h], [\mathbf{X}^l, \hat{\mathbf{X}}^h])$.

### 3.4  Autokrigeability, Complexity, and Further Acceleration

For subset structured data, the computational complexity of GAR is decomposed into two independent TGPs for likelihood and predictive posterior. Thanks to the tensor algebra (mainly $(\mathbf{K} \otimes \mathbf{S})^{-1} = \mathbf{K}^{-1} \otimes \mathbf{S}^{-1}$), the complexity of the $i$-fidelity kernel matrix inversion is reduced to $\mathcal{O}(\sum_{m=1}^M (d_m^i)^3 + (N^i)^3)$ instead of $\mathcal{O}((N^i d^i)^3)$. For the non-subset case, the computational complexity in Eq. (8) is unfortunately $\mathcal{O}((N_m^i d^i)^3)$ where $N_m$ is the number of the imaginary low-fidelity points. Nevertheless, due to the tensor structure, we can still use conjugate gradient [26] to solve the linear system efficiently.

Notice that the mean prediction $\bar{\mathbf{Z}}_*^h$ in Eq. (9) does not depend on any output covariance matrixes $\{\mathbf{S}_m^h, \mathbf{S}_m^l\}_{m=1}^M$, which reassemble the autokrigeability (no knowledge transfer in noiseless cases for mean predictions) [14, 9] based on the GAR framework. For applications where the predictive variation is not of interest, we can introduce a conditional independent output-correlation, i.e., $\mathbf{S}_m^h = \mathbf{I}, \mathbf{S}_m^l = \mathbf{I}$ and orthogonal weight matrixes, i.e., $\mathbf{W}_m^T\mathbf{W}_m = \mathbf{I}$, to reduce the computationally complexity further down to $\mathcal{O}((N^i)^3)$ (see Appendix for detailed proof). We call this CIGAR as an abbreviation for conditional independent GAR. In our empirical assessment, CIGAR is slightly worse than GAR due to the difficulty of ensuring $\mathbf{W}_m^T\mathbf{W}_m = \mathbf{I}$ and numerical noise.

## 4  Related Work

GP for high-dimensional outputs is an important model in many applications such as spatial data modeling and uncertainty quantification. For an excellent review, the readers are referred to [27]. Linear model of coregionalization (LMC) [28, 29] might be the most general framework for high-dimensional GP developed in the geostatistic community. LMC assumes that the full covariance matrix as a sum of constant matrixes timing input-dependent kernels. To reduce model complexity, semiparametric latent factor models (SLFM) [30] simplify LMC by assuming that the matrixes are rank-1 matrixes. Higdon et al. [31] further simplifies SLFM using singular value decomposition (SVD) on the output collection to fix the bases for the rank-1 matrixes. To overcome the linear assumptions of LMC, the (implicit) bases can be constructed in a nonlinear fashion using manifold learning, e.g., KPCA [32] and IsoMap [33] or process convolution [34–36]. Other approaches include multi-task GP, which considers the outputs as dependent tasks [37–39] in a framework similar to LMC and GP regression network (GPRN) [40, 41], which proposes products of GPs to model nonlinear outputs, leading to nontractable models. Despite their success, the complicity of the above approaches are at best $\mathcal{O}(N^3 + d^3)$ whereas some are $\mathcal{O}(N^3 d^3)$, which cannot scale well to high-dimensional outputs for scientific data where $d$ can be, says, 1 million. This problem can be well resolved by introducing tensor algebra [42] to form HOGP [18] or scalable model inference, e.g., in GPRN [43].

Multi-fidelity has become a promising approach to further reduce the data demands in building a surrogate model [13] and Bayesian optimization. The seminal autoregressive (AR) model of Kennedy [3] introduces a linear transformation to univariate high-fidelity outputs. This model was enhanced by Le Gratiet [15] by adopting a deterministic parametric form of linear transformation for the

efficient training scheme as introduced previously. However, it is unclear how AR can deal with high-dimensional outputs. To overcome the linearity of AR, Perdikaris et al. [16] proposes nonlinear AR (NAR). It ignores the output distributions and directly uses the low-fidelity solution as an input for the high-fidelity GP model, which is essentially a concatenating GP structure known as *deep GP* [44]. To propagate uncertainty through the multi-fidelity model, Cutajar et al. [25] uses expensive approximation inference, which makes it prone to overfitting and incapable of dealing with very large dimensional problems. For multi-fidelity Bayesian optimization (MFBO), Poloczek et al. [4] and Kandasamy et al. [45] approximate each fidelity with a GP independently; Zhang et al. [46] use convolution kernel, similar to the process convolution [34, 36] to learn the fidelity correlations; Song et al. [5] combine all fidelity data into one single GP to reduce uncertainty. However, most MFBO surrogates do not scale to high-dimensional problems because they are designed for one target (or at most a couple).

To deal with large dimensional outputs, e.g., spatial-temporal fields, Xing et al. [9] extend AR by assuming a simple additive structure and replacing the simple GPs with scalable multi-output GPs at the cost of losing the power for capturing the output correlations, leading to inferior performance and inaccurate uncertainty estimates; Xing et al. [12] propose Deep coregionalization to extend NAR by learning the latent process [30, 29] extracted from embedding the high-dimensional outputs onto a residual latent space using a proposed residual PCA; Wang et al. [8] further introduce bases propagation along with latent process propagation in a deep GP to increase model flexibility at the cost of significant growth in the number of model parameters and a few simplifications in the approximated inference. Parussini et al. [6] generalize NAR to high-dimensional problems. However, these methods lack a systematic way for joint model training, leading to instability and poor fitting for small datasets. Wu et al. [47] extend GP using neural process to model high-dimensional and non-subset problem effectively. In scientific computing, multi-fidelity fusion has been implemented using stochastic collocation (SC) method [24] for high-dimensional problems, which provides closed-form solutions and efficient design of experiments for the multi-fidelity problem. Xing et al. [7] showed that SC is essentially a special case of AR and proposed active learning to select the best subset for the high-fidelity experiments.

To take the advances of deep learning neural network (NN) and being compatible with arbitrary multi-fidelity data (i.e., non-subset structure), Perrone et al. [22] propose an NN-based multi-task method that can naturally extend to MFBO. Li et al. [23] further extend it as a Bayesian neural network (BNN) to MFBO. Meng and Karniadakis [48] add a physics regularization layer, which requires an explicit form of the problem PDEs, to improve prediction accuracy. To scale for high-dimensional problems with arbitrary dimensions in each fidelity, Li et al. [13] propose a Bayesian network approach to multi-fidelity fusion with active learning techniques for efficiency improvement.

Except for multi-fidelity fusion, AR can be used for model multi-variate problem [49, 50], where GAR can also find its applications. GAR is a general framework for GP-based multi-fidelity fusion of high-dimensional outputs. Specifically, AR is a special case when setting $\mathbf{W} = \rho\mathbf{I}$ and using a separable kernel; ResGP is a special case of GAR by setting $\mathbf{W} = \mathbf{I}$ and $\mathbf{S} = \mathbf{I}$; NAR is a special case of integrating out $\mathbf{W}$ with a normal prior and using a separable kernel; DC is a special case of GAR if it only uses one latent process, integrating out $\mathbf{W}$ as in NAR with a separable kernel; MF-BNN is a finite case of GAR if only one hidden layer is used. See Appendix C for the comparison between SOTA methods.

## 5 Experimental Results

To assess GAR and CIGAR, we compare with the SOTA multi-fidelity fusion methods for high-dimensional outputs including: (1) AR [3], (2) NAR [16], (3) ResGP [9], (4) DC[4] [12], and (5) MF-BNN[5] [13]. All GPs use an RBF kernel for a fair comparison. Because the ARD kernel is separable, the AR and NAR are accelerated using the Kronecker product structure as in GAR for a feasible computation. The original DC with residual PCA cannot deal with unaligned outputs, but it can do so by using an independent PCA, which we called DC-I. Both DCs preserve 99% energy for dimension reductions. MF-BNN is conducted using its default setting. GAR, CIGAR, AR, NAR, and ResGP are implemented using Pytorch[6]. All experiments are run on a workstation with an AMD 5950x CPU and 32 GB RAM.

---

[4] https://github.com/wayXing/DC  [5] https://github.com/shib0li/DNN-MFBO  [6] https://pytorch.org/

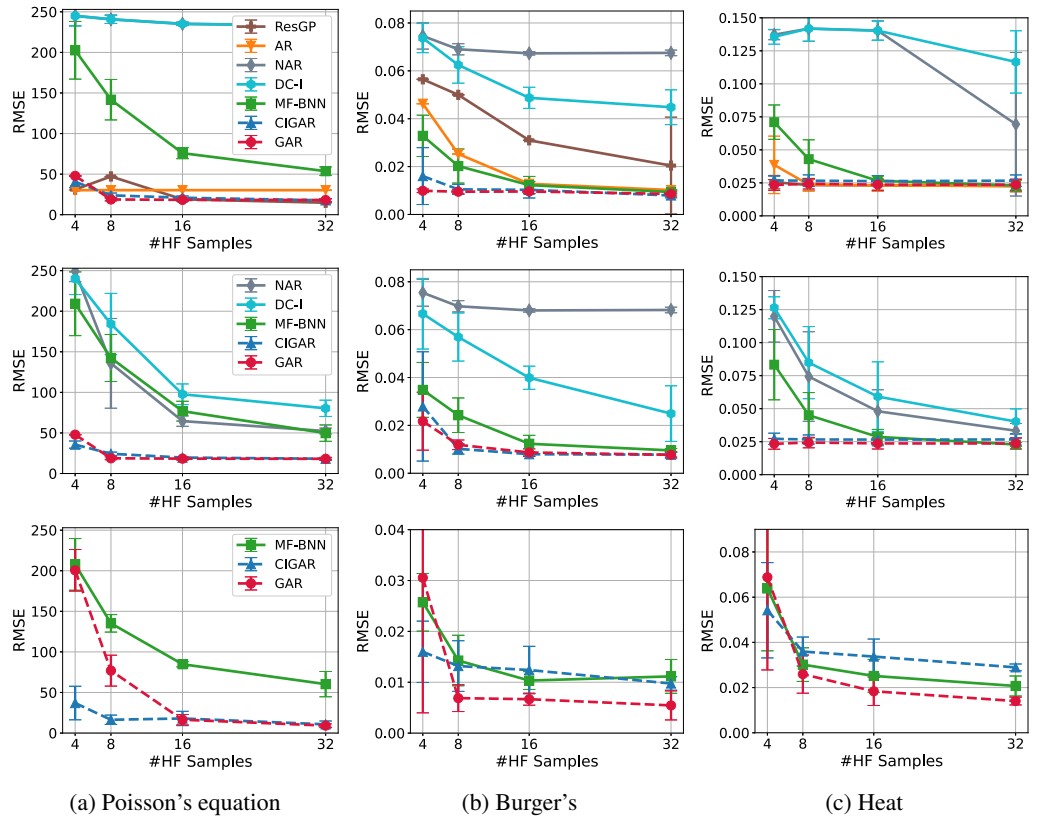

|  |  |  |
|:---:|:---:|:---:|
| (a) Poisson's equation | (b) Burger's | (c) Heat |

Figure 1: RMSE against increasing number of high-fidelity training samples for Poisson's, Burger's, and heat equations with aligned outputs (top row), non-aligned outputs (middle row), and non-subset data (bottom row).

## 5.1 Multi-Fidelity Fusion for Canonical PDEs

We first assess GAR in canonical PDE simulation benchmarks, which produce high-dimensional spatial/spatial-temporal fields as model outputs. Specifically, we test on Burger's, Poisson's and the heat equations commonly used in the literature [12, 51–53]. The high fidelity results are obtained by solving these equations using finite difference on a $32 \times 32$ mesh, whereas the low fidelity by an $8 \times 8$ coarse mesh. The solutions on these grid points are recorded and vectorized as outputs. Because the mesh differs, the dimensionality varies. To compare with the standard multi-fidelity method that can only deal with aligned outputs, we use interpolation to upscale the low-fidelity and record them at the high-fidelity grid nodes. The corresponding inputs are PDE parameters and parameterized initial or boundary conditions. Detailed experimental setups can be found in Appendix E.1

We uniformly generate 128 samples for testing and 32 for training. We increase the high-fidelity training samples to the number of low-fidelity training samples 32. The comparisons are conducted five times with shuffled samples. The statistical results (mean and std) of the RMSE are reported in Fig. 1. GAR and CIGAR outperform the competitors with a significant margin, up to 6x reduction in RMSE and also reaching the optimal performance with a maximum of 8 high-fidelity samples, indicating a successful fusion of low- and high-fidelity. CIGAR is slightly worse than GAR possibility due to the lack of hard constraints on the orthogonality of its weight matrixes during implementation. As we have discovered in the literature, AR consistently performs well. With a flexible linear transformation, GAR outperforms AR while inheriting its robustness, leading to the best performance. For the unaligned output, MF-BNN showed slightly worse performance than in the aligned cases, highlighting the challenges of the unaligned outputs. In contrast, GAR and CIGAR show almost identical performances for both cases. Nevertheless, MF-BNN also shows good performance compared to the rest of the other methods, which is consistent with the finding in [13]. It is interesting to see that for the non-subset data, the capable methods show better performances than in the subset cases. GAR and CIGAR still outperform the competitors with a clear margin.

To approximately assess the performance under an active learning process. We instead generate training samples in a Sobol sequence [54]. The results are shown in Appendix E.2 where GAR and CIGAR also outperform the other methods by a large margin.

## 5.2 Multi-Fidelity Fusion for Real-World Applications

**Optimal topology structure** is the optimized layout of materials, e.g., alloy and concrete, given some design specifications, e.g., external force and angle. This topology optimization is a key technique in mechanical designs, but it is also known for its high computational cost, which renders the need for multi-fidelity fusion. We consider the topology optimization of a cantilever beam with the location of the point load, the angle of the point load, and the filter radius [55] as system inputs. The low-fidelity use a $16 \times 16$ regular mesh for the finite element solver, whereas the high-fidelity $64 \times 64$. Please see Appendix E.3 for a detailed setup.

As in the previous experiment, the RMSE statistics against an increasing number of high-fidelity training samples are shown in Fig. 2. It is clear that GAR outperforms the competitors with a large margin consistently. CIGAR can be as good as GAR when the number of training samples is large.

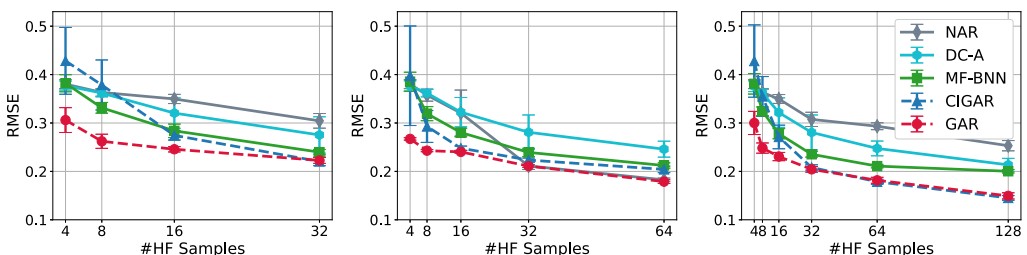

Figure 2: RMSE with low-fidelity training sample number fixed to {32,64,128}.

**Steady-state 3D solid oxide fuel cell**, which solves complex coupled PDEs including Ohm's law, Navier-Stokes equations, Brinkman equation, Maxwell-Stefan diffusion, and convection simultaneously, is a key model for modern fuel cell optimization. The model was solved using finite elements in COMSOL. The inputs were taken to be the electrode porosities, cell voltage, temperature, and pressure in the channels. The low-fidelity experiment is conducted using 3164 elements and relative tolerance of 0.1, whereas the high-fidelity uses 37064 elements and relative tolerance of 0.001. The outputs are the coupled fields of electrolyte current density (ECD) and ionic potential (IP) in the $x - z$ plane located at the center of the channel.

The RMSE statistics are shown in Fig. 5a, which, again, highlights the superiority of the proposed method with only four high-fidelity data training samples. To further assess the model capacity for non-structured outputs, we keep only the ECD (Fig. 5b) and IP (Fig. 5c) in the low-fidelity training data to rise the challenges of predicting high fidelity ECD+IP fields. We can see that removing some low-fidelity information indeed increases the difficulties, especially when removing ECD, where MF-BNN outperforms GAR and CIGAR with a small number of training data. As soon as the training number increases, GAR and CIGAR become superior again.

**Plasmonic nanoparticle arrays** is a complex physical simulation that calculates the extinction and scattering efficiencies $Q_{ext}$ and $Q_{sc}$ for plasmonic systems with varying numbers of scatterers using coupled dipole approximation (CDA), which is a method for mimicking the optical response of an array of similar, non-magnetic metallic nanoparticles with dimensions far smaller than the wavelength of light (here 25 nm). $Q_{ext}$ and $Q_{sc}$ are defined as the QoIs in this work. Please see Appendix E.5 for detailed experiment setup. We conducted the experiments 5 times with shuffled samples, and we fixed the number of low-fidelity training samples to 32, 64, and 128 and gradually increase the high-fidelity training data from 4 to 32, 64, and 128. We can see in Fig.3, GAR outperforms others by a clear margin, especially when the high-fidelity data contains only 4 samples. When there is a large training sample dataset, CIGAR can be as excellent as GAR.

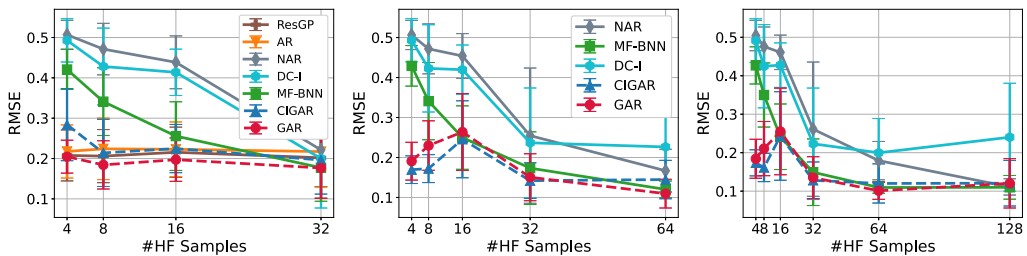

Figure 3: RMSE against an increasing number of high-fidelity training samples with low-fidelity training sample number fixed to {32,64,128} for Plasmonic nanoparticle arrays simulations.

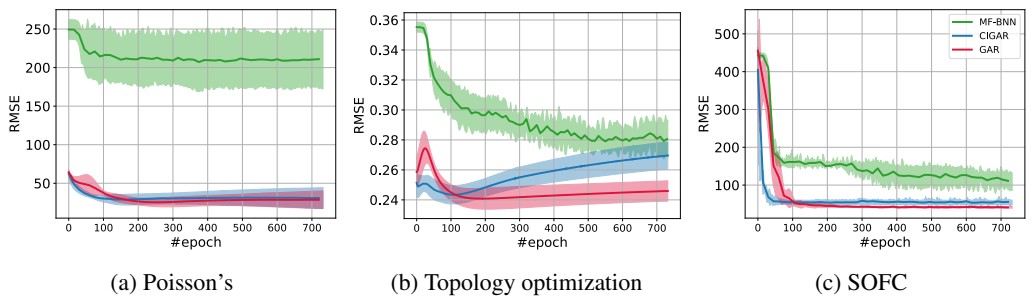

(a) Poisson's      (b) Topology optimization      (c) SOFC

Figure 4: Testing RMSE against an increasing number of training epochs for SOFC, topology optimization, and Poisson datasets.

## 5.3 Stability Test

As a non-parametric model, we expect GAR and CIGAR to have stable performance against overfitting compared to the NN-based methods. In this section, we show the testing RMSE against the training epoch for GAR, CIGAR, and MF-BNN for the previous Poisson's equation, SOFC, and topology optimization. The experiments are repeated five times to ensure fairness. The results are shown in Fig. 4. We can see clearly that GAR and CIGAR are more stable than MF-BNN in almost all cases. The most notables are the converge rate of GAR and CIGAR, which is more than 10x faster if we look at the topology optimization and SOFC cases. For Poisson's equation, the MF-BNN is not likely to match the performance of GAR and CIGAR regardless of the large number of training epochs being used. For the SOFC, MF-BNN, and topology optimization, MF-BNN might be able to match GAR given a very large epoch number, making it a bad choice that consumes expensive computational resources.

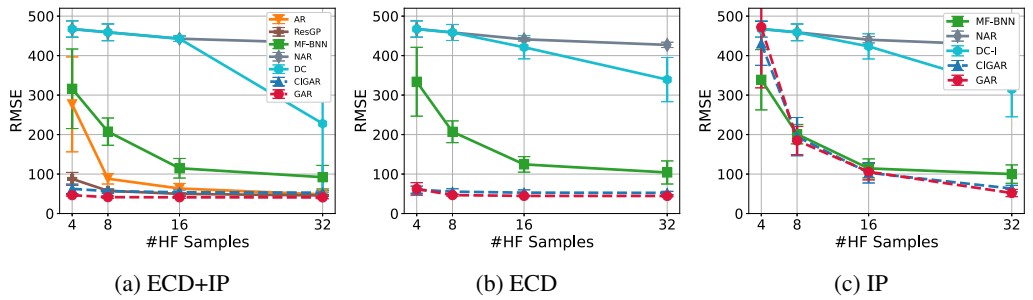

(a) ECD+IP      (b) ECD      (c) IP

Figure 5: RMSE for SOFC with low-fidelity training sample number fixed to 32.

## 6 Conclusion

We propose GAR, the first AR generalization to arbitrary outputs and non-subset multi-fidelity data with a closed-form solution, and CIGAR, an efficient implementation by revealing the autokrigeability in AR. Limitation of this work is scalability w.r.t to samples, lack of active learning [13], and the applications to broader problems of time series and transfer learning[49, 50] using AR-based methods.

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
