# OpenReview forum: "GAR: Generalized Autoregression for Multi-Fidelity Fusion"
_NeurIPS.cc/2022/Conference — NeurIPS 2022 Accept_

### Official Review · Reviewer_QEGg · 2022-07-09

**Rating:** 8
**Confidence:** 3
**Soundness:** 4 excellent
**Presentation:** 4 excellent
**Contribution:** 4 excellent

**Summary:**

The authors proposed Generalized Autoregression (GAR) to handle high-dimensional outputs for multi-fidelity fusion problems. GAR is a GP-based approach that employed tensor algebra to efficiently model covariances across the high dimensions. To handle situations where high fidelity data is not a subset of low fidelity data, they propose a latent variable approach and with a simple trick they proposed CIGAR, which is a computational more efficient version of GAR.

Experiments of GAR and CIGAR showed promising results compared to SOTAs.

**Questions:**

The idea of using imaginary variables to match "missing" data seems very similar to the use of inducing variables in sparse GPs, can you comment a bit on that?

**Limitations:**

Currently only works for Gaussian likelihood. Not sure how straight forward or not it might be to generalised to arbitrary likelihood in high dimensinoal output space.

**Strengths And Weaknesses:**

Strengths:
- clear and thorough comparison of GAR with existing methods in sec.4
- exact solutions are obtainable (when subset data exists)
- although tensor algebra papers tend to be tedious to read, the authors defined the notations clearly and communicated effectively between equations.
- the problem is clearly defined, and worthwhile contribution to handle high dim output multi fidelity fusion problem.

Weakness:
- there are some typos around, e.g. line 38 there is a 1. after citation [9], or in line 55 the word "Problems" is capitalised, but these do not affect the reading experience at all.
- seems to only work for regression problems for now - if non-gaussian likelihood is needed, not sure how straight forward it is to translate the whole tensor algebra business to that. Worth mentioning in the paper a bit as well I guess? Nonetheless, focusing on regression data is a worthwhile contribution already in my opinion.

---

> ### Author Response · Authors · 2022-08-02
> **Thank you and resolving questions**
>
> Thank you very much for your valuable feedback on our work.
>
> C1: seems to only work for regression problems for now - if non-gaussian likelihood is needed, not sure how straight forward it is to translate the whole tensor algebra business to that. Worth mentioning in the paper a bit as well I guess? Nonetheless, focusing on regression data is a worthwhile contribution already in my opinion.
>
> R1: Yes, currently, our model only deals with regression problems only. This is mainly due to the fact that almost all previous works are designed for regression problems. It is kind of rare to see real applications with multi-fidelity non-gaussian likelihood data. Thus, we follow the literature to focus on the regression problem.
> We think the reviewer has proposed an exciting direction for the multi-fidelity problem. There can be a non-gaussian likelihood for regression, e.g. if we apply a Box-Cox Transformation. Nevertheless, there are already many existing tensors-based models for non-gaussian likelihood for us to follow [1,2]. We will certainly extend our work for non-gaussian likelihood in the future. Thank you for your very insightful advice.
>
> [1] Xu, Zenglin, and Feng Yan. "Infinite Tucker decomposition: Nonparametric Bayesian models for multiway data analysis." arXiv preprint arXiv:1108.6296 (2011).
>
> [2] Xing, Wei, et al. "Infinite ShapeOdds: Nonparametric Bayesian Models for Shape Representations." Proceedings of the AAAI Conference on Artificial Intelligence. Vol. 34. No. 04. 2020.
>
> C2: The idea of using imaginary variables to match "missing" data seems very similar to the use of inducing variables in sparse GPs, can you comment a bit on that?
>
> R2: Yes. If we leave the tensor part aside, they are indeed sparse GPs. When we introduce the tensor, each output corresponds to two parts: one is the known input parameters, which form a normal GP, and the other part corresponds to the “inducing variables” that form a sparse GP (with inducing points).
>
>
> In summary, we appreciate the positive feedback from the reviewer. We genuinely believe that we are submitting a solid work that is fundamental and novel to the multi-fidelity fusion community. We will also open source our code to benefit potential users. Please let us know if you find any things unclear or require our further efforts to improve this work. Thank you!

---

### Official Review · Reviewer_UYDR · 2022-07-10

**Rating:** 6
**Confidence:** 3
**Soundness:** 3 good
**Presentation:** 3 good
**Contribution:** 3 good

**Summary:**

The authors here present a generalized autoregressive model to conduct multi-fidelity prediction on data with high-dimensional outputs via tensor decomposition. A different non-subset multi-fidelity data setting is explored. Based on autokrigeability, a more efficient implementation is proposed.

**Questions:**

1. How do you decide the size of the tensor?

**Strengths And Weaknesses:**

Strength:
1)	Problems are clearly stated and addressed.
2)	The non-subset data scenario is addressed here for the first time.
3)	Tensor GP is applied here to reflect possible underlying structures of the data.
Weakness:
1)	The notations are a bit confusing, like Z, F and their subscripts.
2)	Sparse variational inference might further improve the inference efficiency here.

---

> ### Author Response · Authors · 2022-08-02
> **Thank you and resolving questions**
>
> Thank you for your valuable suggestions for our work.
>
> C1: Sparse variational inference might further improve the inference efficiency here.
>
> R1: We agree with the reviewer. Indeed this is our plan for the next step to improve this method for a very large number of data points.
>
> C2 How do you decide the size of the tensor?
>
> R2: For this work, we keep its original tensor format because each spatial-temporal data admits a natural tensor structure. For instance, for a spatial-temporal field with input $x_i$ in a 2d-space domain, we use a 4-mod tensor $(\xi, x_1, x_2,t)$ to index the tensor, where $x$ indicates the space and $t$ indicates the time.
>
> For data without an explicit tensor format, we follow the HOGP work [1] to organize it into a random tensor. This will not create issues because the index is then learned by the model. However, the indexes are not likely to have the same meaning to its original indexes.
>
> We genuinely believe that we are submitting a solid work that is fundamental and novel to the multi-fidelity fusion community. We will also open source our code to benefit potential users. Please let us know if you find any things unclear or require our further efforts to improve this work (and its rating :) ). Thank you!

---

### Official Review · Reviewer_ebFJ · 2022-07-12

**Rating:** 5
**Confidence:** 4
**Soundness:** 3 good
**Presentation:** 3 good
**Contribution:** 2 fair

**Summary:**

This paper designs a generalized autoregression method (GAR) and its efficient implementation CIGAR to deal with non-structured high-dimensional outputs and non-subset multi-fidelity data. The result shows their method outperforms the baselines among several benchmarks at the highest fidelity level.

**Questions:**

The paper mentioned uncertainty quantification. But why there's no evaluation of uncertainty quantification for experiments? It will be great if the author could include other metrics for accuracy (negative log-likelihood) and uncertainty quantification (Continuous Ranked Probability Score [1] or mean interval score [1,2]).

[1] Gneiting, Tilmann, and Adrian E. Raftery. "Strictly proper scoring rules, prediction, and estimation." Journal of the American statistical Association 102.477 (2007): 359-378.
[2] Wu, Dongxia, et al. "Quantifying uncertainty in deep spatiotemporal forecasting." arXiv preprint arXiv:2105.11982 (2021).


**Limitations:**

No potential negative societal impact I can see.

**Strengths And Weaknesses:**

Strengths:
Well-written paper in general.
Include 5 experiments. 2 with real-world data.

Weaknesses:
My main concern about the paper is its novelty. Although the author didn't overclaim their contributions "Generalization of the AR for arbitrary non-structured high-dimensional outputs" and "Generalization to non-subset multi-fidelity data for the AR", which are all for the classic autoregression model specifically. But there's existing non-AR work [1] that is able to handle both non-structured high-dimensional outputs and non-subset multi-fidelity data [1] besides MF-BNN.

[1] Wu, Dongxia, et al. "Multi-fidelity Hierarchical Neural Processes." arXiv preprint arXiv:2206.04872 (2022).

Besides, there's no big improvement between the proposed method and baselines (MF-BNN, ResGP) for some of the experiments. Only one metric RMSE is evaluated.

---

> ### Author Response · Authors · 2022-08-02
> **Thank you, adding experiments, and resolving questions**
>
> Thank you for your valuable suggestions for our work.
>
> C1: But there's existing non-AR work [1] that is able to handle both non-structured high-dimensional outputs and non-subset multi-fidelity data [1] besides MF-BNN.
>
> R1: Thank you for suggesting this great work. We have cited this work in our revision. As it is presented in the introduction (and is also pointed out by the suggested work [1]), AR is a dominant technique for multi-fidelity fusion. Our work aims to generalize AR to resolve its critical limitation for high-dimension and non-subset challenges. This way, we can derive a simple yet powerful model to deal with most multi-fidelity problems. Unlike many existing deep learning-based methods with thousands of model parameters (such as MF-BNN), the proposed method has only a few model parameters (less than 10 hyperparameters for CIGAR) and thus can naturally deal with a small dataset, which is essentially important for multi-fidelity fusion in the emulation of physics simulations. The superiority of the proposed method is well supported by our experiments, where GAR/CIGAR outperforms the SOTA methods in most cases. We will also compare our results with the latest great work [1] in the future.
>
> C2: Besides, there's no big improvement between the proposed method and baselines (MF-BNN, ResGP) for some of the experiments. Only one metric RMSE is evaluated.
>
> R2: We believe that the proposed method has significantly improved the SOTA methods, including MF-BNN and ResGP. There are indeed situations (non-subset data for Poisson’s, Burger’s, and heat equations with 4 training data points) where the proposed method is no better than MF-BNN. However, as the number of training data points increases to 8 and more, our method outperforms other models with a significant margin. We can confidently conclude that our method shows improved performance for 95% of all cases we have tested (including 6 different datasets), which itself sufficiently justifies the superiority.
>
> The significance of our work is also partially recognized by the recommended reference [1], where AR is recognized as a predominant method but lacks the ability to deal with high-dimension and non-subset situations, the exact issues we resolve in this work.
>
> C3: The paper mentioned uncertainty quantification. But why there's no evaluation of uncertainty quantification for experiments? It will be great if the author could include other metrics for accuracy (negative log-likelihood) and uncertainty quantification (Continuous Ranked Probability Score [1] or mean interval score [1,2]).
>
> R3: We agree with the reviewer. Unfortunately, due to the limited timeframe, we are not able to conduct a comprehensive investigation with all suggested metrics. We instead add the most commonly used log-likelihood (also used by MF-HNP [1]) as a fair comparison for the real-world datasets. The detailed results and discussions have been supplemented in Appendix E.7. They are consistent with the previous conclusions. Note that we cannot add the results for MF-BNN, because the open-source code does not provide a clear way to generate the predictive variance, and the published codes are difficult to modify without discussions with their authors. We will contact its authors and hopefully resolve this problem in the future.
>
> We would like to bring up that most multi-fidelity literature (even GP-based method) still uses RMSE as the main metric (e.g. in MF-BNN, ResGP, DC, NAR ) to compare different methods. Being able to outperform other SOTA methods under RMSE consistently is a significant contribution.
>
> Also, we hope the reviewer can see the significance of our work and not underrate it for that it generalizes the predominant multi-fidelity fusion technique, AR, for the modern high-dimension and non-subset challenges. The elegance and tractable framework of AR allows GAR to be further enhanced and modified for different challenging problems, e.g., non-Gaussian likelihood.
>
> We genuinely believe that we are submitting a solid work that is fundamental and novel to the multi-fidelity fusion community. We will also open source our code to benefit potential users. Please let us know if you find any things unclear or require our further efforts to improve this work (and your rating :) ). Thank you!

---

> > ### Comment · Reviewer_ebFJ · 2022-08-07
> > **Response**
> >
> > Thanks for answering my questions. I've improved my score based on the responses.

---

### Meta-Review · Area_Chair_nDTu · 2022-08-27

**Recommendation:** Accept
**Confidence:** Certain

**Metareview:**

This paper considers the problem of multi-fidelity fusion using generalized autoregression. The authors especially take on problems such as high-dimensionality and non-subsetness with this approach. The reviewers agree that the paper is well written and makes a significant contribution to MF-fusion. I recommend acceptance and strongly encourage the authors to take the reviewer comments into account in preparing the final manuscript.

**Award:**

No

---

### Decision · Program_Chairs · 2022-09-14

Accept